environmental engineering/analysis

orthogonal experiment, river sediment, ceramsite, Beian River

**Author for correspondence:**
Hongzhi Ma
e-mail: mahongzhi@ustb.edu.cn

# Ceramsite production from sediment in Beian River: characterization and parameter optimization

Yong Jin[1,2], Songyu Huang[3], Qunhui Wang[1,2], Ming Gao[1,2] and Hongzhi Ma[1,2,4]

[1]Department of Environmental Engineering, University of Science and Technology, Beijing 100083, People's Republic of China
[2]Beijing Key Laboratory of Resource-oriented Treatment of Industrial Pollutants, Beijing 100083, People's Republic of China
[3]Research Institute of Enfi, China Enfi Engineering Corporation, 12 Fuxing Avenue, Beijing 100083, People's Republic of China
[4]Tianjin Sunenergy Sega Environmental Science and Technology Co. Ltd, Tianjin 300380, People's Republic of China

HM, 0000-0001-5968-3115

In order to realize pollution control and resource recovery, sediment from Beian River in Mudanjiang City China was used for ceramsite production. The maximum content of total nitrogen (TN), total phosphorus (TP) and organic matter (OM) in sediments of Beian River were 2975 mg kg$^{-1}$, 2947 mg kg$^{-1}$ and 29.6%, respectively. So, it should be treated properly for resource utilization. The orthogonal experiment of $L_{16}$ ($4^5$) was adopted to determine the best conditions for ceramsite production and the result demonstrated that the sewage sludge ratio of 15%, binder ratio of 5%, pre-heating temperature of 450°C, sintering temperature of 1150°C and firing time of 23 min were the optimum conditions. The corresponding product met with the standard of CJ/T 299-2008 and the heavy metal leaching experiment showed it was lower than the threshold of China's industrial standard. Thus, it demonstrated that ceramsite production was a feasible way for utilization of sediment.

## 1. Introduction

The sediment is a loose aggregate of sand, clay, residue and other materials deposited at the bottom of the river [1]. It is an important part of the multi-phase ecosystem of water. The pollution status of sediment is an important factor to measure the quality of water environment comprehensively [2,3]. So, the

management of huge amount of sediments coming from dredging activities in harbours or channels is an important issue to be solved in many countries. To maintain the navigability of the waterways, several 100 million tons of sediments are dredged around the world each year [4].

For many years, dredged sediments are considered waste materials, and they are mainly landfilled as the slurry. Due to the shortage of disposal capacity, nowadays, many reuse strategies have attracted attention, and dredging sediment is increasingly regarded as a resource [5,6]. A possible application for the recycled dredged materials is for manufacturing building materials such as bricks, cement clinker and lightweight aggregate [7,8]. Lafhaj *et al.* [9] stabilized the sediment in the northern French river by the *Novosol* method, and the amount of sediment was 0–45%. It was found that the compressive strength of the brick increased with the amount of sediment. Besides, the leaching test showed that the leaching rate of heavy metals was within the specified limits and would not cause harm to the environment. Dalton *et al.* [10] used dredged sediment to produce cement, and conducted laboratory and pilot tests. The results by X-ray diffractometer showed that the sintering temperature mainly depended on the content of quartz in the sediment.

Besides, the ceramsite production has become a research hot point gradually. Ceramsite, with high porosity, large specific surface area, low bulk, apparent density and low toxicity, has been widely used as building materials and water treatment filter media [11]. And there is some research on different sediments to produce ceramsite, such as clay, shale and other natural resources. Wang & Xu [12] used the Suzhou River sediment, fly ash and iron powder as additives to produce building ceramic aggregate, the results showed that the product performance fully met the requirements of the technical indicators in the national standard of clay ceramsite [13]. Tay *et al.* [14] used industrial sludge and dredged sediment to produce ceramsite. The compressive strength of concrete made from the ceramsite could reach $31–39\,N\,mm^{-2}$, which could be used as sewage treatment after improvement. Liu & Xi [15] used river sediment, sewage sludge, Guangxi white mud and water glass as additives to produce a new type of water treatment ceramsite. It showed that different sediments could produce ceramsite with different properties, and these ceramsites could be used in different ways, such as building materials, filter materials, etc.

The Beian River in Mudanjiang city is in the north of the Mudanjiang railway. It is formed by three tributaries and receives about 90 000 tons of sewage per day, and it had a natural runoff of $0.45\,m^3\,s^{-1}$. Due to the discharge of a large amount of untreated industrial wastewater and sewage, the Beian River and its tributaries have become polluted rivers and affect the surrounding environment. Although the Mudanjiang municipal government carried out sediment dredging and ecological restoration to control the river pollution, it was also a question that how to deal with the large amount of sludge generated by dredging.

The purpose of this study was to investigate the properties of ceramsite from Beian River sediment. In this study, the optimum conditions of ceramsite production were simulated and optimized by orthogonal experiment for the first time from five factors: sludge content, binder content, pre-heating time, sintering temperature and firing time. The removal of total nitrogen (TN), total phosphorus (TP), polycyclic aromatic hydrocarbons (PAHs) and heavy metals from ceramsite was also investigated. The method of parameter optimization provides a new angle and research method for the resource utilization of river sediment.

# 2. Material and methods

## 2.1. Sampling positions

The sludge in different positions of Beian River was sampled for analysis and further utilization. Taking into account the different positions of river in different seasons, three positions, which demonstrated the upstream, middle and downstream of the river, were chosen for the sampling. The details are shown in figure 1.

(1) The first location was chosen as 30 m downstream at the confluence of the two rivers (Jinlong Xi and Yinlong Xi), which came from the rural area of the city, this sampling point represented the characteristics of sediment for domestic water and did not include industrial sewage.
(2) The second point was 50 m downstream of Beian bridge, mainly in the residential and commercial area, so it represented the pollutions came from these areas.

... 

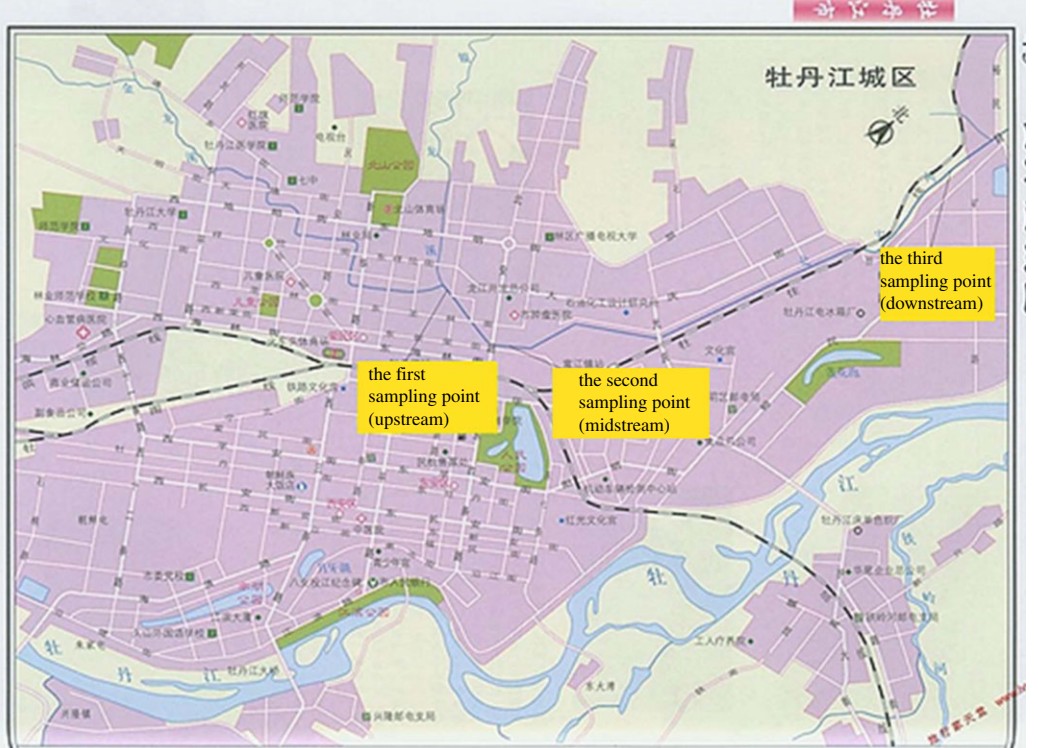

**Figure 1.** Diagram of sediment sampling in Beian River.

(3) The third point was 100 m upstream of the confluence of Beian River and Mudan River. This point was in the industrial zones; on both sides of the river, there were a lot of industrial companies. This sampling point represented the industrial pollution to the sediment.

## 2.2. Methods

### 2.2.1. Pretreatment for the sediment and sludge

The sediments were sampled and then brought to the laboratory for analysis and further preparation. The collected samples were air-dried, crushed, ground, mixed and passed through a 100-mesh sieve.

The dehydrated sewage sludge was taken from the sludge dewatering workshop of the one wastewater treatment plant; and the moisture content was 70–80%. It was transported to the test site and dried, crushed and sieved for further use.

### 2.2.2. Production process

The process of preparing the sediment ceramsite in the laboratory is shown in figure 2. The whole process can be divided into five stages: raw material pretreatment, moulding, drying, sintering and cooling.

The raw materials were stirred in a dry powder mixed for 10 min and pelletized to particle sizes of 0.6–0.8 cm and then left in draught cupboard at 200°C for 1 h. After drying treatment, raw pellets held in a porcelain crucible were preheated in a muffle oven and then were rapidly shifted into an electric tube furnace (sintered at 1150°C for 23 min). Finally, the product was obtained after cooling down to the room temperature.

### 2.2.3. Analytical methods

The total nitrogen (TN) of the sediment was determined by potassium persulfate digestion ultraviolet spectrophotometry [16]; total phosphorus (TP) was determined using the molybdenum blue method with an ultraviolet spectrophotometer (UV754N) at a wavelength of 254 nm. $NH_4Cl$ and $KH_2PO_4$ were of analytical reagent grade [17]; and organic matter (OM) was determined by potassium dichromate

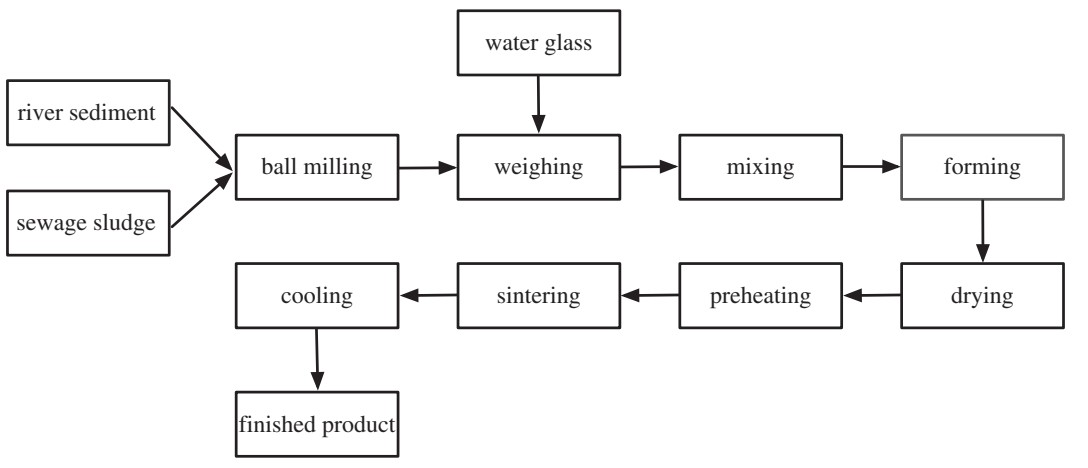

**Figure 2.** Process flow for preparation of ceramic pellets from river sediments.

**Table 1.** Orthogonal experimental factors table. All the data were based on dry materials.

| level | unit | 1 | 2 | 3 | 4 |
|---|---|---|---|---|---|
| sewage sludge | % | 0 | 5 | 10 | 15 |
| binder | % | 0 | 5 | 10 | 15 |
| pre-heating temperature | ℃ | 350 | 400 | 450 | 500 |
| sintering temperature | ℃ | 1120 | 1140 | 1160 | 1180 |
| firing time | min | 15 | 18 | 20 | 23 |

method [18]. PAHs in sediment were mainly determined by accelerated solvent extraction, chromatography column purification and gas chromatography–mass spectrometry (GC/MS) [19].

The Brunauer–Emmett–Teller (BET) procedure of nitrogen adsorption was used to determine the specific surface area of ceramsite filter media [20]. The formula is as follows:

$$\frac{P/P_0}{V(1 - P/P_0)} = \frac{C-1}{V_m C} \times \frac{P}{P_0} + \frac{1}{V_m C},$$ (2.1)

where $P$ is the equilibrium adsorption pressure (Pa), $P_0$ is the saturated steam pressure (Pa), $V$ is the adsorption volume in standard state (cm$^3$), $V_m$ is the single-layer adsorption volume in standard state (cm$^3$) and $C$ is the BET constant.

The X-ray fluorescence spectrometry was used to analyse the composition of samples. The national standard 'Artificial Ceramsite Filter for Water Treatment' was used to measure the bulk density [21]. The inductively coupled plasma was used for the determination of heavy metals in ceramsite leaching solution.

## 2.3. Experimental design

### 2.3.1. Orthogonal experiment

In order to determine the best experimental condition, the experiment of five factors and four levels was selected for orthogonal experiment. $L_{16} 4^5$ is used as shown in table 1; the sewage sludge ratio, binder ratio, pre-heating temperature, sintering temperature and firing time were selected.

Specific surface area and loose bulk density were chosen as the evaluation indicators for the ceramsite. According to the test table, the results showed that there was an inverse correlation between the two evaluation indexes, that is, the smaller the loose bulk density, the better the test effect; while the larger the specific surface area, the better the test effect. Thus, comprehensive index was adopted in this study:

$$\text{comprehensive index} = \frac{\text{specific surface area}}{\text{loose bulk density}} \times 1000.$$

**Table 2.** Determination of main chemical composition of sediments in the Beian River. All the data were based on dry materials.

| chemical composition | general ceramic raw material | the Beian River |
|---|---|---|
| $SiO_2$ (%) | 48–70 | $55.8 \pm 6$ |
| $Al_2O_3$ (%) | 15–25 | $17.5 \pm 0.8$ |
| $Fe_2O_3$ (%) | 3–12 | $4.86 \pm 0.22$ |
| CaO + MgO (%) | 1–12 | $1.62 \pm 0.09$ |
| $K_2O + Na_2O$ (%) | 2.5–7 | $4.53 \pm 0.1$ |
| loss of ignition | | $8.3 \pm 0.24$ |

**Table 3.** Main chemical composition of dehydrated sewage sludge samples. All the data were based on dry materials.

| chemical composition | $SiO_2$ | $Al_2O_3$ | $Fe_2O_3$ | CaO | MgO | $K_2O$ | $Na_2O$ | loss of ignition |
|---|---|---|---|---|---|---|---|---|
| content % | 8.96 | 1.45 | 1.62 | 3.09 | 0.65 | 0.12 | 0.08 | 67.28 |
| metal | Cu | Zn | Cr | Pb | Cd | Fe | | |
| content mg kg$^{-1}$ | 1185.6 | 120.2 | 85.6 | 160.5 | 15.5 | 875.2 | | |

As an evaluation index for the evaluation of orthogonal test results, the larger the comprehensive index, the better the test results.

### 2.3.2. Heavy metal leaching experiment

In order to investigate the environment safety for the ceramsite produced in this study, heavy metal leaching test was performed with the national standard [22]. At first, weighed 100 g of sample and place it in a polyethylene bottle. Second, added the leachate with the ratio of mud to water 1 : 10 (w/v, g ml$^{-1}$), and set the constant temperature water bath oscillator [(110 ± 10) r min$^{-1}$]; the leaching time was 8 h. Then the filtrate was collected as the leaching solution and performed analysis, the average value of three measurements was used for discussion.

# 3. Results and discussion

## 3.1. Analysis of sediment properties

### 3.1.1. Chemical composition

The main chemical compositions in the sediment raw materials played a very important role during the firing process. Therefore, it must be determined in detail before firing the ceramsite. These materials mainly included quartz, $Al_2O_3$, calcite, alkali metal and organic.

The chemical composition of the Beian River sediment was analysed in detail in the laboratory, and the main metal oxide content was measured and compared with the chemical composition of the ceramsite raw materials (table 2).

From table 2, we can see that only the content of CaO + MgO was slightly lower, but the basic chemical composition of the Beian River sediment completely met the requirements of the raw materials of the ceramsite.

The main chemical components and heavy metal content of the sewage sludge samples are determined as shown in table 3.

It can be seen from table 3 that the sewage sludge also had a certain heavy metal content. Therefore, in order to prevent the prepared ceramsite products from causing secondary pollution, we need to strengthen the consolidation of heavy metals in the preparation process of the sediment ceramsite.

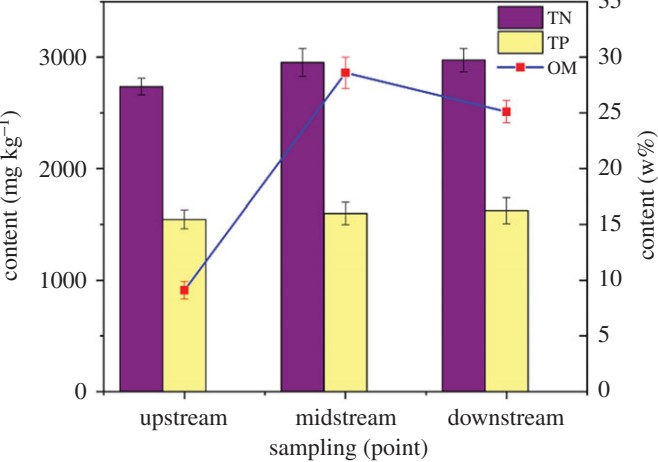

**Figure 3.** Sampling variation of TN, TP and OM in dry season.

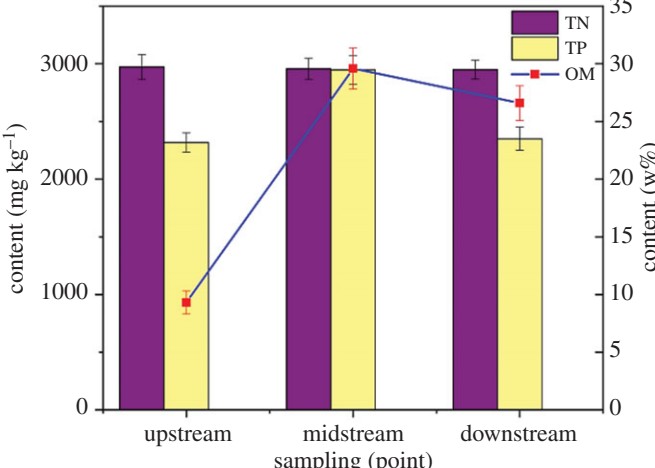

**Figure 4.** Sampling variation of TN, TP and OM in wet season.

### 3.1.2. Determination of TN, TP and OM contents

Nitrogen, phosphorus and OM pollutants can be enriched in the sediment after long-term accumulation. At the same time, they can be released into the water and cause secondary pollution. Therefore, the measurement of TN, TP and OM content can determine the degree of contamination of sediments. What is more, the content of OM is the main basis for sediment dredging. In this study, TN, TP and OM in sediments of different periods and watersheds of Beian River were determined according to the test methods. The average values of sediments were measured many times at the same place. The results are shown in figures 3 and 4.

At present, in order to eliminate endogenous pollution of nitrogen and phosphorus in sediment dredging, the dredging standards of TN > 1000 mg l$^{-1}$ and TP > 500 mg l$^{-1}$ were proposed by Ni *et al.* [23].

Through the analysis of figures 3 and 4, it can be seen that the nitrogen and phosphorus contents of sediments sampled in the Beian River during the wet and dry season were seriously over the standard. The TN content in the wet season was slightly higher than dry season, and it was little changed in different watersheds, while the TN content in the midstream and downstream was slightly higher than the upstream during the dry season. This might be due to the uniform distribution of nitrogen-containing substances in the water during the wet season, and the accumulation of nitrogen in the midstream and downstream during the dry season.

The TP content in the midstream was higher than that in the upstream and downstream during the wet season, and there was little difference in the TP content in different watersheds during the dry

season. This might be because with the increase in water during the wet season, the TP index increased as the phosphorus-containing substances entered the water. The midstream were mainly living and commercial areas, so the phosphorus content was higher than other streams. It could be founded that sediment dredging of Beian River must be carried out.

According to the determination results in figures 3 and 4, the OM content in the sediment of the Beian River during dry and wet seasons were similar, and it was low in the upstream. This was because the water speed was faster, which led to the scouring of sediment and was not easy to silt up, so OM content was low or mainly sandstone; the OM content in the midstream was the highest, because the midstream was concentrated. In living and commercial areas, a large amount of domestic refuse was washed into the Beian River, which resulted in the long-term transformation of most OM and accumulated in the river. In the downstream, although the OM content was lower than the midstream, it was also significantly higher. This was due to the existence of some industrial enterprises and industrial wastewater was discharged from both sides, so foreign pollution sources contributed greatly to the increase in OM content in sediment.

The content of OM in the sediment of the midstream and downstream of Beian River was about 10%. By comparison, it could be found that the contents of OM in the sediment of the midstream and downstream of Beian River were much higher than other rivers. At the same time, considering the low water content of the sediment in Beian River, the sediment had a good application prospect in the utilization of dredged sediment.

### 3.1.3. Determination of PAHs pollutants

PAHs, produced by incomplete combustion of coal, petroleum, wood, tobacco and organic macromolecule compounds, are volatile hydrocarbons which pollute the environment and food seriously [24]. Based on the investigation of Beian River in different seasons and watersheds for a long time, and considering the characteristics of Beian River being polluted by different ways, the types of PAHs in different seasons and watersheds of Beian River were determined according to the test methods. The results are shown in electronic supplementary material, table S1.

In electronic supplementary material, table S1, PAHs are marked with Slash font. According to the results of measurement, it could be seen that there were many kinds of PAHs in the sediments, whether in dry or wet season. On the other hand, it verified that the Beian River had been polluted by many kinds of pollution such as living, commercial and industrial pollution for a long time. The results show that the PAHs in the midstream and downstream of Beian River were obviously more than those in the upstream. This might be due to the serious commercial and industrial pollution in the midstream and downstream of Beian River, which proved that Beian River was a heavily polluted river with multiple pollution effects.

## 3.2. Parameter optimization by orthogonal experiments

According to the factors and levels in table 1, orthogonal experiments were carried out according to table 4.

In this study, according to the evaluation with the loose bulk density and specific surface area, the best test condition was $A_4B_2C_3D_3E_4$. They were sewage sludge ratio of 15%, binder ratio of 5%, pre-heating temperature of 450°C, sintering temperature of 1150°C and the firing time of 23 min. Based on specific surface area, bulk density and particle density, Han *et al*. [25] designed a four-factors and three-levels orthogonal experiment to further find that the best quality ratio of sludge to auxiliary materials was sludge : fly ash : clay = 2 : 3 : 1. The optimum process conditions for firing ceramsite were: drying time 1 h, pre-heating temperature 300°C, pre-heating time 20 min and the sintering temperature 1100°C, the sintering time 8 min. Finally, the sludge was used as the main raw material, fly ash and clay as auxiliary materials, and the ceramsite was obtained with good performance. But, there was something different between these two studies. On the one hand, our study used more inorganic matter, but Han used more OM. On the other hand, our sintering temperature was higher than theirs. At 1100°C, the ceramsite could form a continuous skeleton and a well-closed pore structure. But, the densification was extremely high at 1150°C. Microscopical porous structure with a diameter size of about 10.0 μm was formed, which have a promising application for using as filter media in sewage treatment [26].

After comprehensive evaluation and analysis, it showed that the importance order for the factors was: sewage sludge > sintering temperature > firing time > binder > pre-heating temperature. Zhang & Xi [27] found that sewage sludge, Guangxi white mud, binder, sintering temperature and firing time had

**Table 4.** Orthogonal experimental results. All the data were based on dry materials.

| factor | A sewage sludge (%) | B binder (%) | C pre-heating temperature (°C) | D sintering temperature (°C) | E firing time (min) | assessment index loss weight (kg m$^{-3}$) | specific surface area (m$^2$g$^{-1}$) | comprehensive index specific surface area/loose bulk density × 1000 |
|---|---|---|---|---|---|---|---|---|
| 1 | 1 | 1 | 1 | 1 | 1 | 720.12 | 2.63 | 3.65 |
| 2 | 1 | 2 | 2 | 2 | 2 | 685.45 | 2.91 | 4.25 |
| 3 | 1 | 3 | 3 | 3 | 3 | 645.27 | 2.87 | 4.45 |
| 4 | 1 | 4 | 4 | 4 | 4 | 608.23 | 2.02 | 3.32 |
| 5 | 2 | 1 | 2 | 3 | 4 | 642.11 | 2.89 | 4.50 |
| 6 | 2 | 2 | 1 | 4 | 3 | 741.02 | 3.13 | 4.22 |
| 7 | 2 | 3 | 4 | 1 | 2 | 692.78 | 1.98 | 2.86 |
| 8 | 2 | 4 | 3 | 2 | 1 | 564.43 | 2.12 | 3.76 |
| 9 | 3 | 1 | 3 | 4 | 2 | 789.09 | 2.65 | 3.36 |
| 10 | 3 | 2 | 4 | 3 | 1 | 661.28 | 3.42 | 5.17 |
| 11 | 3 | 3 | 1 | 2 | 4 | 649.56 | 2.56 | 3.94 |
| 12 | 3 | 4 | 2 | 1 | 3 | 556.21 | 1.96 | 3.52 |
| 13 | 4 | 1 | 4 | 2 | 3 | 655.43 | 3.07 | 4.68 |
| 14 | 4 | 2 | 3 | 1 | 4 | 610.26 | 3.65 | 5.98 |
| 15 | 4 | 3 | 2 | 4 | 1 | 615.2 | 2.46 | 4.00 |
| 16 | 4 | 4 | 1 | 3 | 2 | 583.09 | 2.39 | 4.10 |
| k1 | 3.92 | 4.05 | 3.98 | 4.00 | 4.15 | | | |
| k2 | 3.84 | 4.12 | 4.07 | 4.12 | 3.63 | | | |
| k3 | 4.00 | 3.81 | 4.39 | 4.56 | 4.22 | | | |
| k4 | 4.69 | 3.66 | 4.01 | 3.72 | 4.44 | | | |
| R | 0.85 | 0.46 | 0.41 | 0.84 | 0.81 | | | |

**Table 5.** Heavy metal leaching comparison table (mg l$^{-1}$). All the data were based on dry materials.

| | Cu | Zn | Cr | Pb | Cd |
|---|---|---|---|---|---|
| raw material | 1.28 | 1.86 | 0.131 | 0.183 | 0.045 |
| ceramsite | 0.16 | 0.25 | 0.022 | 0.004 | 0.001 |
| standard | 50 | 50 | 10 | 3 | 0.3 |

significant effects on the test results through orthogonal test and analysis of variance. The order of significance was sewage sludge > sintering temperature > Guangxi white mud > firing time > binder. It could be found that the sewage sludge was the most important factor, because the sewage sludge could increase the specific surface area, decrease the bulk density and the sintering temperature.

After determining the optimal formulation of the experiment ($A_4B_2C_3D_3E_4$) by orthogonal experiment, the corresponding performance of the product was measured according to the ceramsite industry standard promulgated by the Ministry of Construction. The results are in electronic supplementary material, table S2. And it could be found that the ceramsite prepared from Beian River sediment was in full compliance with the relevant industry standards promulgated by the Ministry of Construction.

## 3.3. Analysis of heavy metal leaching of ceramsite

The leaching toxicity test could effectively simulate the release of heavy metals in solids when they were in contact with water. Each country has developed specific guidelines, mainly based on a chemical approach, for characterizing dredged material referring to different regulatory agencies (e.g. US EPA and European agencies) [28]. In order to characterize the environmental impacts of the geocomposites obtained with dredged sediments, Lirer et al. [4] performed the European standard leaching tests. To determine the effectiveness of immobilization and the mechanism of leaching of contaminants from controlled low-strength material forms, Zhen et al. [29] performed leaching tests by using the toxicity characteristic leaching procedure (US EPA Test Method 1311-TCLP).

Due to the complex composition of the sediment samples and the fact that the contaminated sediment samples contain a certain amount of heavy metal composition, secondary pollution of heavy metals may occur during the application process after resource utilization. So, we need to use the standard method [30] to investigate the leaching toxicity of ceramsite. The results we measured are shown in table 5.

According to the data analysis in table 5, it could be seen that although the heavy metal pollution of the Beian River sediment was serious, the total amount of heavy metal leaching did not exceed the standard, which indirectly verifies that the heavy metals in the sediment of Beian River were stable. The heavy metal leaching amount of the ceramsite was much smaller than those in the raw materials. The reason may be that some heavy metals, such as chromium, cadmium, copper, zinc and lead, interacted with silicate or aluminosilicate matrix at high temperature [31]. The solidifying efficiencies of the heavy metals were enhanced strongly by the crystallization and chemical incorporation within the aluminosilicate or silicate frameworks during thermal treatment, and the heavy metals available for leaching were decreased significantly.

At the same time, the ceramsite samples with better firing effect were measured for basic indexes such as specific surface area, bulk density and water absorption, and compared with the relevant standards of artificial filter materials for water treatment in the Ministry of Construction (electronic supplementary material, table S3). According to the comparative analysis of electronic supplementary material, table S3, it could be seen that the ceramsite prepared from the Beian River sediment was in full compliance with the relevant industry standards promulgated by the Ministry of Construction, and many indicators were much higher than this industry standard, so we could determine that the programme was feasible.

## 4. Conclusion

The ceramsite production from sediments in Beian River mixed with sewage sludge was studied with the analysis of component and safety consideration. The sediments were used for clay substitution. The orthogonal experiments with sewage sludge content, binder content, sintering temperature, firing time

and pre-heating temperature as main influencing factors were optimized in this study. In this way, sustainable development for the sediment of the Beian River was acquired.

Data accessibility. Data available from the Dryad Digital Repository: https://doi.org/10.5061/dryad.5j0h14j [32].

Authors' contributions. H.M., Y.J. and M.G. designed the study. Y.J. and S.H. performed the experiment, Y.J., S.H. and M.G. analysed the data. H.M., Y.J. and S.H. wrote the manuscript. Q.W. and M.G. gave the suggestion for the experiment and helped to revise the manuscript. S.H. prepared the sample for analysis. All authors gave final approval for publication.

Competing interests. We declare we have no competing interests.

Funding. The work was supported by Major Science and Technology Program for Water Pollution Control and Treatment (2012ZX07201002-6), National Scientific Funding of China (no. 51378003) and the Fundamental Research Funds for the Central Universities (FRF-BD-17-014A).

Acknowledgements. We are grateful to Ming Gao and Qunhui Wang, who provided suggestions during the experiment, and we thank Songyu Huang for her assistance with the sample analysis.

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
