## [Reviewer comments · Royal Society Open Science]

Review History

RSOS-190197.R0 (Original submission)

Review form: Reviewer 1

Is the manuscript scientifically sound in its present form?

Yes

Are the interpretations and conclusions justified by the results?

Yes

Is the language acceptable?

Yes

Is it clear how to access all supporting data?

Yes

Do you have any ethical concerns with this paper?

No

Have you any concerns about statistical analyses in this paper?

No

Recommendation?

Accept with minor revision (please list in comments)

Comments to the Author(s)

The article utilized the sediments from Beian River in North China to produce construction materials and also the properties for the sediments were characterized. The orthogonal experimental design was adopted to determine the optimum conditions and corresponding evaluation methods were also performed. I think the paper could be considered for publication with the following issues resolved:

- 1) As the writer indicated, the sediments contained some PAHs, particularly for the sediments in downside of the river, during the treatment will these materials decrease completely?
- 2) The sediments were used for ceramsite production, is this the best resource technology, or is there any other alternative for the utilization of the sediment?
- 3) The sample position is in north China, will the location affect the property and utilization of the sediments?

Review form: Reviewer 2

Is the manuscript scientifically sound in its present form?

Yes

Are the interpretations and conclusions justified by the results?

Yes

Is the language acceptable?

Yes

Is it clear how to access all supporting data?

Not Applicable

Do you have any ethical concerns with this paper?

No

Have you any concerns about statistical analyses in this paper?

No

Recommendation?

Accept with minor revision (please list in comments)

Comments to the Author(s)

The sediments in the river are of great importance for environmental safety of the river. More and more research demonstrated that the proper treatment of the sediments would reduce the pollution source for the river. This paper investigated the pollution characteristic of the sediments and carried out research on resource technology. The paper contained the following

problems, after the revision of paper it could be considered for publication in the journal.

- 1) The author mentioned several technologies for sediments utilization, why ceramsite production was adopted in this study?
- 2) What would be the total volume of the sediments in this area, did the author think of the economic feasibility of the process?
- 3) The author investigated the product quality of the product and showed the environmental safety of the product, so will the quality of the product fulfill its utilization requirement?

Review form: Reviewer 3 (Jian Ouyang)

Is the manuscript scientifically sound in its present form?

Yes

Are the interpretations and conclusions justified by the results?

Yes

Is the language acceptable?

Yes

Is it clear how to access all supporting data?

Yes

Do you have any ethical concerns with this paper?

No

Have you any concerns about statistical analyses in this paper?

I do not feel qualified to assess the statistics

Recommendation?

Accept with minor revision (please list in comments)

Comments to the Author(s)

Good paper. This study is interesting and meaningful to sediment disposal. There are a few comments to the authors before publication.

1. This study focused on using the sediment in Beian River to produce ceramsite. Is it suitable for other sediment in other rivers? It will be very meaningful if the authors can illustrate which sediment in river is suitable for produce ceramsite.
2. The conclusion seems like a summary of the work. The authors should state the new technical or scientific findings in here.
3. Some sentences are difficult to be read, such as the first paragraph in the section 2.2.3 and the second paragraph in the section 2.3.1.

Decision letter (RSOS-190197.R0)

04-Jun-2019

Dear Dr Ma,

The editors assigned to your paper ("Ceramsite Production from Sediment in Beian River: Characterization and Parameter Optimization") have now received comments from reviewers. We would like you to revise your paper in accordance with the referee and Associate Editor suggestions which can be found below (not including confidential reports to the Editor). Please note this decision does not guarantee eventual acceptance.

Please submit a copy of your revised paper before 27-Jun-2019. Please note that the revision deadline will expire at 00.00am on this date. If we do not hear from you within this time then it will be assumed that the paper has been withdrawn. In exceptional circumstances, extensions may be possible if agreed with the Editorial Office in advance. We do not allow multiple rounds of revision so we urge you to make every effort to fully address all of the comments at this stage. If deemed necessary by the Editors, your manuscript will be sent back to one or more of the original reviewers for assessment. If the original reviewers are not available, we may invite new reviewers.

- Data accessibility

If you wish to submit your supporting data or code to Dryad (<http://datadryad.org/>), or modify your current submission to dryad, please use the following link:
<http://datadryad.org/submit?journalID=RSOS&manu=RSOS-190197>

- **Competing interests**

- **Authors' contributions**

- **Acknowledgements**

- **Funding statement**

Kind regards,

Alice Power

Editorial Coordinator

on behalf of R. Kerry Rowe (Subject Editor)

Comments to Author:

Reviewers' Comments to Author:

Reviewer: 1

Comments to the Author(s)

The article utilized the sediments from Beian River in North China to produce construction materials and also the properties for the sediments were characterized. The orthogonal experimental design was adopted to determine the optimum conditions and corresponding evaluation methods were also performed. I think the paper could be considered for publication with the following issues resolved:

- 1) As the writer indicated, the sediments contained some PAHs, particularly for the sediments in downside of the river, during the treatment will these materials decrease completely?
- 2) The sediments were used for ceramsite production, is this the best resource technology, or is there any other alternative for the utilization of the sediment?
- 3) The sample position is in north China, will the location affect the property and utilization of the sediments?

Reviewer: 2

Comments to the Author(s)

The sediments in the river are of great importance for environmental safety of the river. More and more research demonstrated that the proper treatment of the sediments would reduce the pollution source for the river. This paper investigated the pollution characteristic of the sediments and carried out research on resource technology. The paper contained the following problems, after the revision of paper it could be considered for publication in the journal.

- 1) The author mentioned several technologies for sediments utilization, why ceramsite production was adopted in this study?
- 2) What would be the total volume of the sediments in this area, did the author think of the economic feasibility of the process?
- 3) The author investigated the product quality of the product and showed the environmental safety of the product, so will the quality of the product fulfill its utilization requirement?

Reviewer: 3

Comments to the Author(s)

Good paper. This study is interesting and meaningful to sediment disposal. There are a few comments to the authors before publication.

1. This study focused on using the sediment in Beian River to produce ceramsite. Is it suitable for other sediment in other rivers? It will be very meaningful if the authors can illustrate which sediment in river is suitable for produce ceramsite.
2. The conclusion seems like a summary of the work. The authors should state the new technical or scientific findings in here.
3. Some sentences are difficult to be read, such as the first paragraph in the section 2.2.3 and the second paragraph in the section 2.3.1.

Author's Response to Decision Letter for (RSOS-190197.R0)

See Appendix A.

RSOS-190197.R1 (Revision)

Review form: Reviewer 1

Is the manuscript scientifically sound in its present form?

Yes

Are the interpretations and conclusions justified by the results?

Yes

Is the language acceptable?

Yes

Do you have any ethical concerns with this paper?

No

Have you any concerns about statistical analyses in this paper?

No

Recommendation?

Accept as is

Comments to the Author(s)

Authors have revised manuscript according to reviewers' comments. I recommend that the revised manuscript can be accepted to publish in the journal.

Review form: Reviewer 2

Is the manuscript scientifically sound in its present form?

Yes

Are the interpretations and conclusions justified by the results?

Yes

Is the language acceptable?

Yes

Do you have any ethical concerns with this paper?

No

Have you any concerns about statistical analyses in this paper?

No

Recommendation?

Accept as is

Comments to the Author(s)

Accept.

Review form: Reviewer 3 (Jian Ouyang)

Is the manuscript scientifically sound in its present form?

Yes

Are the interpretations and conclusions justified by the results?

Yes

Is the language acceptable?

Yes

Do you have any ethical concerns with this paper?

No

Have you any concerns about statistical analyses in this paper?

No

Recommendation?

Accept as is

Comments to the Author(s)

The revision can be satisfied now.

Decision letter (RSOS-190197.R1)

11-Jul-2019

Dear Dr Ma,

I am pleased to inform you that your manuscript entitled "Ceramsite Production from Sediment in Beian River: Characterization and Parameter Optimization" is now accepted for publication in Royal Society Open Science.

on behalf of Prof R. Kerry Rowe (Subject Editor)
openscience@royalsociety.org

Reviewer comments to Author:

Reviewer: 1

Comments to the Author(s)

Authors have revised manuscript according to reviewers' comments. I recommend that the revised manuscript can be accepted to publish in the journal.

Reviewer: 2

Comments to the Author(s)

Accept.

Reviewer: 3

Comments to the Author(s)

The revision can be satisfied now.

Appendix A

Thanks for the reviewers' suggestions and remarks on our manuscript. Our replies are as follows.

(Replies to questions of reviewers in Blue words; Revisions in red words)

Corresponding corrections were made in the text in red.

Reviewer's suggestions or questions	Original text	Replies or revisions in revised manuscript
Answer to Referee 1		
1 As the writer indicated, the sediments contained some PAHs, particularly for the sediments in downside of the river, during the treatment will these materials decrease completely?		Answer: Thanks for the suggestion. In the process of making ceramsite from Beian River sediment, the sintering temperature reached 1150°C, and PAHs were completely removed.
2 The sediments were used for ceramsite production, is this the best resource technology, or is there any other alternative for the utilization of the sediment?		Answer: Thanks for the suggestion. Ceramsite production is one kind of resource technology for sediments utilization, it is effective particularly for the sediments with high inorganic concentration and needed thermal treatment to remove the potential hazardous materials. Other methods of utilizing river sediment are brick making, biochar and biofilm materials.
3 The sample position is in north China; will the location affect the property and utilization of the sediments?		Answer: Thanks for the suggestion. The composition of river sediment in different areas will be different, generally the river in north China is seasonal river, Beian river is such kind of river, it has a fast flow speed in wet season, and the sediments has more inorganic components, while the river in south China

		generally will not be frozen, and sometimes organic components will be easily accumulated in the sediments. For the sediments with different composition, different resource technology should be adopted to realize resource utilization and value-added product.
Answer to Referee 2		
1 The author mentioned several technologies for sediments utilization, why ceramsite production was adopted in this study?		Answer: Thanks for the suggestion. It was found that the sediment of Beian River contained high concentration of inorganic materials and shared the similarity with those of clay, that was the main reason for utilization ceramsite production, furthermore the thermal treatment during the sintering process could also reduce the organic compounds in the sediments and increase its utilization safety.
2 What would be the total volume of the sediments in this area, did the author think of the economic feasibility of the process?		Answer: Thanks for the suggestion. The total volume of sediment in this area is 37500m³ (10*0.5*7500), the river has an average length of 7500 m, the sediments has an average depth of 0.5 m and the width of river is about 10 m. The thermal treatment for the sediment to produce ceramsite is an energy consumption process. in our process, we also need the sewage sludge and water glass as the raw materials to produce the product, we just

		provided a technology for waste utilization, the economical feasibility needed further investigation, since pretreatment is also needed in this process, also the process water should be considered.
3 The author investigated the product quality of the product and showed the environmental safety of the product, so will the quality of the product fulfill its utilization requirement?	The results were in supplementary table (Table S2). And it could be found that the ceramsite prepared from Beian River sediment was in full compliance with the relevant industry standards promulgated by the Ministry of Construction.	Answer: Thanks for the suggestion. From the text, we can find that the ceramsite prepared from Beian River sediment fulfill its utilization requirement, the industry standards promulgated by the Ministry of Construction of China. And also, the environmental safety was checked by the experiment in the study.
Answer to Referee 3		
1 This study focused on using the sediment in Beian River to produce ceramsite. Is it suitable for other sediment in other rivers? It will be very meaningful if the authors can illustrate which sediment in river is suitable for produce ceramsite.	The sediment is a loose aggregate of sand, clay, residue and other materials deposited at the bottom of the river ^[1] . Although the Mudanjiang municipal government carried out sediment dredging and ecological restoration to control the river pollution, it was also a question that how to deal with the large amount of sludge generated by dredging.	Answer: Thanks for the suggestion. The purpose of this study is to produce ceramsite from the sediment of Beian River and provide theoretical basis for the utilization of other river sediments. But it is different among the climate, sediment composition, and pollution degree, so it will be investigated in the future. The sediment is mainly a loose aggregate of sand, clay, and residue deposited at the bottom of river. And the main materials we used for ceramsite production were dredging sludge.
2 The conclusion seems like a summary of the work. The authors should state the new technical or scientific findings in here.	In this study, the sediment properties of Beian River were sampled, determined and analyzed. The high concentration of TN and	Answer: Thanks for the suggestion. I have revised it in the text. The ceramsite production from sediments in Beian River

	TP demonstrated it should be treated properly. Then the ceramsite production from sediments in Beian River mixed with sewage sludge was studied. The orthogonal experiments with sewage sludge content, binder content, sintering temperature, firing time and pre-heating temperature as main influencing factors were optimized. And the further detect for the product showed it meet the construction standard and showed environmental safety. In this way, sustainable development for the sediment of the Beian River was acquired.	mixed with sewage sludge was utilized with the analysis of component and safety consideration. The sediments were used for clay substation, the orthogonal experiments with sewage sludge content, binder content, sintering temperature, firing time and pre-heating temperature as main influencing factors were optimized in this study. In this way, sustainable development for the sediment of the Beian River was acquired.
3 Some sentences are difficult to be read, such as the first paragraph in the section 2.2.3 and the second paragraph in the section 2.3.1.	 ● The total nitrogen of the sediment was determined by potassium persulfate digestion ultraviolet spectrophotometry ^[16]; total phosphorus by potassium persulfate digestion ammonium molybdate spectrophotometry ^[17]; and organic matter by potassium dichromate method ^[18]. Polycyclic aromatic hydrocarbons (PAHs) in sediment were mainly determined by accelerated solvent extraction (ASE), chromatography column purification and gas chromatography-mass spectrometry (GC/MS) ^[19]. ● Specific surface area and loose bulk density were 	Answer: Thanks for the suggestion. We have made revision in the text.  ● The total nitrogen (TN) of the sediment was determined by potassium persulfate digestion ultraviolet spectrophotometry ^[16]; Total phosphorus (TP) was determined using the molybdenum blue method with an ultraviolet spectrophotometer (UV754N) at a wavelength of 254 nm. NH₄Cl and KH₂PO₄ were of analytical reagent grade ^[17]; and organic matter (OM) was determined by potassium dichromate method ^[18]. Polycyclic aromatic hydrocarbons (PAHs) in

	chosen as the evaluation indicators for the ceramsite. According to the test table, the results showed that two evaluation indexes were contradictory, that is, the smaller the loose bulk density, the better the test effect; while the larger the specific surface area, the better the test effect. Thus, comprehensive index was adopted in this study:	sediment were mainly determined by accelerated solvent extraction (ASE) , chromatography column purification and gas chromatography-mass spectrometry (GC/MS) [19]. ● Specific surface area and loose bulk density were chosen as the evaluation indicators for the ceramsite. According to the test table, the results showed that two evaluation indexes were contradictory there was an inverse correlation between the two evaluation indexes, that is, the smaller the loose bulk density, the better the test effect; while the larger the specific surface area, the better the test effect. Thus, comprehensive index was adopted in this study:
--	---	--